# Diagnostic accuracy of a novel tuberculosis point-of-care urine lipoarabinomannan assay for people living with HIV: A meta-analysis of individual in- and outpatient data

Tobias Broger[1*], Mark P. Nicol[2,3,4*], Rita Székely[1], Stephanie Bjerrum[5,6], Bianca Sossen[7,8], Charlotte Schutz[7,8], Japheth A. Opintan[9], Isik S. Johansen[5,6], Satoshi Mitarai[10], Kinuyo Chikamatsu[10], Andrew D. Kerkhoff[11], Aurélien Macé[1], Stefano Ongarello[1], Graeme Meintjes[7,8], Claudia M. Denkinger[1,12‡], Samuel G. Schumacher[1‡*]

1 FIND, Geneva, Switzerland, 2 Division of Infection and Immunity, School of Biomedical Sciences, University of Western Australia, Perth, Western Australia, Australia, 3 Division of Medical Microbiology, University of Cape Town, Cape Town, South Africa, 4 National Health Laboratory Service, Cape Town, South Africa, 5 Mycobacterial Research Centre of Southern Denmark, Department of Infectious Diseases, Odense University Hospital, Odense, Denmark, 6 Department of Clinical Research, Unit of Infectious Diseases, University of Southern Denmark, Odense, Denmark, 7 Department of Medicine, Faculty of Health Sciences, University of Cape Town, Cape Town, South Africa, 8 Wellcome Center for Infectious Diseases Research in Africa, Institute of Infectious Disease and Molecular Medicine, University of Cape Town, Cape Town, South Africa, 9 Department of Medical Microbiology, School of Biomedical and Allied Sciences, College of Health Sciences, University of Ghana, Accra, Ghana, 10 Department of Mycobacterium Reference and Research, Research Institute of Tuberculosis, Japan Anti-Tuberculosis Association, Tokyo, Japan, 11 Division of HIV, Infectious Diseases and Global Medicine, Zuckerberg San Francisco General Hospital and Trauma Center, Department of Medicine, University of California, San Francisco, California, United States of America, 12 Division of Tropical Medicine, Center for Infectious Diseases, Heidelberg University Hospital, Heidelberg, Germany

☉ These authors contributed equally to this work.
‡ These authors are joint senior authors on this work.
* Samuel.Schumacher@finddx.org

## Abstract

### Background

Tuberculosis (TB) is the most common cause of death in people living with HIV (PLHIV), yet TB often goes undiagnosed since many patients are not able to produce a sputum specimen, and traditional diagnostics are costly or unavailable. A novel, rapid lateral flow assay, Fujifilm SILVAMP TB LAM (SILVAMP-LAM), detects the presence of TB lipoarabinomannan (LAM) in urine, and is substantially more sensitive for diagnosing TB in PLHIV than an earlier LAM assay (Alere Determine TB LAM lateral flow assay [LF-LAM]). Here, we present an individual participant data meta-analysis of the diagnostic accuracy of SILVAMP-LAM in adult PLHIV, including both published and unpublished data.

### Methods and findings

Adult PLHIV (≥18 years) were assessed in 5 prospective cohort studies in South Africa (3 cohorts), Vietnam, and Ghana, carried out during 2012 to 2017. Of the 1,595 PLHIV who

**Data Availability Statement:** All relevant data are within the manuscript and its Supporting Information files.

**Funding:** This work was funded by the Global Health Innovative Technology (GHIT) Fund (grant number G2017-207), the UK Department for International Development (DFID) (grant number 300341-102), the Dutch Ministry of Foreign Affairs (grant number PDP15CH14), the Bill & Melinda Gates Foundation (grant number OPP1105925), the Australian Department of Foreign Affairs and Trade (grant number 70957) and the German Federal Ministry of Education and Research (BMBF; grant number 2020 62 156). The Cohort2 study was funded by the Wellcome Trust (088590 and 085251). GM was supported by Wellcome Trust (098316 and 203135/Z/16/Z), the South African Research Chairs Initiative of the Department of Science and Technology and the National Research Foundation (NRF) of South Africa (Grant No 64787), NRF incentive funding (UID: 85858) and the South African Medical Research Council through its TB and HIV Collaborating Centres Programme, with funds received from the National Department of Health (RFA#SAMRC-RFA-CC:TB/HIV/AIDS-01-2014). CS received funding from the South African Medical Research Council through the National Health Scholarship Programme. ADK received funding from the National Institute of Allergy and Infectious Diseases (Grant No T32 AI060530). BS received salary support from the Wellcome Trust (grant number 088316). CMD is supported by a fellowship of the Burroughs–Wellcome Fund from the American Society of Tropical Medicine and Hygiene. The opinions, findings and conclusions expressed in this manuscript reflect those of the authors alone. The funders had no role in study design, data collection and analysis, decision to publish, or preparation of the manuscript.

**Competing interests:** I have read the journal's policy and the authors of this manuscript have the following competing interests: TB, SGS, AM, SO, RS and CMD were previously or are currently employed by FIND. TB reports a patent in the field of lipoarabinomannan detection. CMD is a member of *PLOS Medicine*'s Editorial Board. The rest of the authors declare no competing interests associated with this manuscript. The corresponding author had full access to all the data in the study and had final responsibility for the decision to submit for publication.

**Abbreviations:** CRS, composite reference standard; IPD, individual patient data; LAM, lipoarabinomannan; LF-LAM, Alere Determine TB LAM lateral flow assay; M.tb, *Mycobacterium*

met eligibility criteria, the majority (61%) were inpatients, median age was 37 years (IQR 30–43), 43% had a CD4 count ≤ 100 cells/μl, and 35% were receiving antiretroviral therapy. Most participants (94%) had a positive WHO symptom screen for TB on enrollment, and 45% were diagnosed with microbiologically confirmed TB, using mycobacterial culture or Xpert MTB/RIF testing of sputum, urine, or blood. Previously published data from inpatients were combined with unpublished data from outpatients. Biobanked urine samples were tested, using blinded double reading, with SILVAMP-LAM and LF-LAM. Applying a microbiological reference standard for assessment of sensitivity, the overall sensitivity for TB detection was 70.7% (95% CI 59.0%–80.8%) for SILVAMP-LAM compared to 34.9% (95% CI 19.5%–50.9%) for LF-LAM. Using a composite reference standard (which included patients with both microbiologically confirmed as well as clinically diagnosed TB), SILVAMP-LAM sensitivity was 65.8% (95% CI 55.9%–74.6%), and that of LF-LAM 31.4% (95% CI 19.1%–43.7%). In patients with CD4 count ≤ 100 cells/μl, SILVAMP-LAM sensitivity was 87.1% (95% CI 79.3%–93.6%), compared to 56.0% (95% CI 43.9%–64.9%) for LF-LAM. In patients with CD4 count 101–200 cells/μl, SILVAMP-LAM sensitivity was 62.7% (95% CI 52.4%–71.9%), compared to 25.3% (95% CI 15.8%–34.9%) for LF-LAM. In those with CD4 count > 200 cells/μl, SILVAMP-LAM sensitivity was 43.9% (95% CI 34.3%–53.9%), compared to 10.9% (95% CI 5.2%–18.4%) for LF-LAM. Using a microbiological reference standard, the specificity of SILVAMP-LAM was 90.9% (95% CI 87.2%–93.7%), and that of LF-LAM 95.3% (95% CI 92.2%–97.7%). Limitations of this study include the use of biobanked, rather than fresh urine samples, and testing by skilled laboratory technicians in research laboratories, rather than at the point of care.

## Conclusions

In this study, we found that SILVAMP-LAM identified a substantially higher proportion of TB patients in PLHIV than LF-LAM. The sensitivity of SILVAMP-LAM was highest in patients with CD4 count ≤ 100 cells/μl. Further work is needed to demonstrate accuracy when implemented as a point-of-care test.

## Author summary

### Why was this study done?

- Tuberculosis (TB) is the most common cause of death in people living with HIV (PLHIV); however, TB is difficult to diagnose in PLHIV because patients may have extrapulmonary disease, difficulty producing a sputum sample, or few TB bacilli in their sputum.

- Rapid point-of-care urine testing for TB with the Alere Determine TB LAM lateral flow assay (LF-LAM) reduces mortality in patients with advanced HIV disease, but LF-LAM has only moderate sensitivity, and its uptake in countries with high burdens of TB has been slow.

- Several recent studies have shown that the Fujifilm SILVAMP TB LAM (SILVAMP-LAM) test has improved sensitivity over LF-LAM and comparable specificity in PLHIV.

*tuberculosis*; MRS, microbiological reference standard; NTM, nontuberculous mycobacteria; PLHIV, people living with HIV; POC, point-of-care; SILVAMP-LAM, Fujifilm SILVAMP TB LAM; TB, tuberculosis; Xpert, Xpert MTB/RIF.

## What did the researchers do and find?

- We did an individual patient meta-analysis of the accuracy of SILVAMP-LAM across 5 cohort studies in South Africa, Ghana, and Vietnam, and compared SILVAMP-LAM results with those of LF-LAM.

- SILVAMP-LAM was twice as sensitive as LF-LAM in detecting TB in PLHIV, irrespective of whether a reference standard of microbiologically proven or clinically diagnosed TB was used. There were more apparent false-positive results associated with SILVAMP-LAM than with LF-LAM, although this difference between the 2 tests was small (4.4 percentage point difference in specificity).

## What do these findings mean?

- SILVAMP-LAM is a promising new rapid urine test for TB in PLHIV, particularly in those with advanced HIV disease, who are at highest risk of death.

- Further work is needed to determine whether SILVAMP-LAM, which is slightly more complex to perform than LF-LAM, can be reliably done at the point of care, and what impact the implementation of SILVAMP-LAM has on mortality in PLHIV with TB.

See S1 Translation for the Japanese language Abstract.

## Introduction

Tuberculosis (TB) is the most common cause of death in people living with HIV (PLHIV), whose risk of developing active TB is estimated to be approximately 30 times greater than in people without HIV [1]. Evidence from randomized diagnostic studies shows that early diagnosis of TB among PLHIV reduces mortality [2,3].

Traditional diagnostic methods, such as culture or smear microscopy, are slow or low in sensitivity. More sensitive modern techniques, such as Xpert MTB/RIF (Xpert), require a certain infrastructure, are costly, and are not widely accessible. Moreover, TB is harder to diagnose in PLHIV, since many of the patients have extrapulmonary TB (approximately 25%), produce paucibacillary sputum samples, or cannot reliably produce a sputum specimen [4,5]. TB in PLHIV is often fatal if undiagnosed or left untreated. New, rapid, non-sputum-based point-of-care (POC) diagnostic solutions to detect TB, especially in vulnerable groups, are urgently needed [6].

The commercially available Alere Determine TB LAM lateral flow assay (LF-LAM; Abbott, Chicago, US; in previous studies also called AlereLAM) is a rapid, inexpensive POC TB test [7]. While its use is associated with a mortality benefit in severely ill and immunocompromised PLHIV [2,3], it has only moderate sensitivity in patients with a low CD4 count and has had low programmatic uptake [8–10]. We have already reported on the novel Fujifilm SILVAMP TB LAM (SILVAMP-LAM; Fujifilm, Tokyo, Japan; in previous studies also called FujiLAM) assay. SILVAMP-LAM, similarly to LF-LAM, detects the presence of lipoarabinomannan (LAM) in urine using a visually read lateral flow test, but unlike LF-LAM, SILVAMP-LAM utilizes silver amplification. For inpatients with HIV, it offers on average an increase in sensitivity

of approximately 25%–30% compared to LF-LAM across CD4 strata, while maintaining a high specificity when a composite reference standard is used [11].

Here, we present all available individual patient data (IPD), both published [11,12] and unpublished, from SILVAMP-LAM testing ("intervention" per PRISMA guidelines [13]) in adult inpatients and outpatients with HIV ("patients") in comparison to LF-LAM ("comparator") across 5 prospective cohorts ("studies") and analyze its diagnostic accuracy ("outcome"). These data contributed to a guidance development group consultation of the World Health Organization in May 2019.

## Methods

### Study population

Biobanked urine samples from adult PLHIV (≥18 years), collected in 5 prospective cohort studies in South Africa, Vietnam, and Ghana, were assessed (S1 Table). Study protocols and statistical analysis plans for the different studies are available upon request. A study that combines inpatient cohorts from South Africa (cohorts 1A, 2, and 3) [11] and the study from Ghana (cohort 5) [12] have been published.

For the first cohort from South Africa, adults with TB symptoms able to produce sputum were enrolled consecutively, independently of their HIV status. Inpatients were enrolled on admission to Khayelitsha Hospital (cohort 1A), while outpatients were enrolled at the Town Two and Nolungile primary healthcare facilities in the Khayelitsha township (cohort 1B), between February 2017 and August 2017. Those in whom the disease was thought to be only extrapulmonary were excluded.

The second cohort from South Africa (cohort 2) enrolled adult inpatients with HIV consecutively, independently of their CD4 count, as they were admitted to adult medical wards at GF Jooste Hospital between June 2012 and October 2013, regardless of their ability to produce sputum or whether they reported TB symptoms [14].

The third cohort from South Africa (cohort 3) enrolled inpatient PLHIV at Khayelitsha Hospital with a CD4 ≤ 350 cells/μl in whom TB was considered the most likely diagnosis at presentation between January 2014 and October 2016 [15]. A list of all potentially eligible patients was compiled daily, and a random selection procedure (using a die after all potentially eligible patients had been identified) was followed to enroll 2–4 patients daily.

The fourth cohort (cohort 4), from Vietnam, used samples from consecutively enrolled patients presenting to the outpatient clinics of a public sector district hospital (Pham Ngoc Thach Hospital, Ho Chi Minh City) with symptoms suggestive of TB, independently of HIV status, between September 2016 and July 2017.

For cohort 5, HIV-infected adults eligible for antiretroviral therapy were consecutively enrolled irrespective of whether they reported TB signs and symptoms from an outpatient clinic in the Fevers Unit of Korle-Bu Teaching Hospital, a public referral hospital in Accra, the capital city of Ghana, between January 2013 and March 2014 [12,16].

All cohorts excluded patients who were already receiving anti-TB therapy. Where patients were enrolled independently of HIV status, the present study included only PLHIV. More details on the individual cohorts can be found in S1 Table and S2 Table. Relevant data are available in S1 Data. Study protocols are available upon request.

All study-related activities were approved by the human research ethics committees of the respective sites including City of Cape Town (Ref. 10364a) and the University of Cape Town Human Research Ethics Committee (UCT HREC, Ref. 250/2018) for cohort 1, UCT HREC (Ref. 001/2012) for cohort 2, UCT HREC (Ref. 057/2013) for cohort 3, Ministry of Health Vietnam (Ref. 2493/QĐ-BYT) for cohort 4, and the Institutional Review Board of University of

Ghana Medical School (Ref. MS-Et/M.4-P3.3/2012-13) and the Danish National Committee (Ref. 1302133/Doc No. 1206169) for cohort 5. Written informed consent was obtained from patients, as per the study protocols. Study participation did not affect standard of care. All reporting follows STARD and PRISMA guidelines [17,18] (S1 STARD Checklist and S1 PRISMA IPD Checklist), and no analysis plan was prespecified.

## Laboratory methods

**Samples.** Urine specimens were stored at −80˚C (cohort 1, cohort 3, cohort 4) and −20˚C (cohort 2, cohort 5) (S3 Table). Aliquots of frozen, unprocessed urine were thawed to ambient temperature and mixed manually prior to testing with SILVAMP-LAM and LF-LAM. Samples that were not immediately used for testing were stored at 4˚C for a maximum of 4 hours.

**Users and training.** SILVAMP-LAM and LF-LAM testing was performed by skilled laboratory technicians in research laboratories at the University of Cape Town and the Research Institute of Tuberculosis of the Japan Anti-Tuberculosis Association. The technicians were newly trained to perform SILVAMP-LAM. Training included review of the English instructions for use, explanation and demonstration of both assays by a trainer, and conductance of up to 3 tests by the users followed by a proficiency test questionnaire. The user's ability to correctly interpret the LF-LAM band intensity was assessed using 20 scanned LF-LAM test strips with different results next to the 4-grade reference scale card. The total training time for both tests was 4–6 hours.

**Index testing (SILVAMP-LAM).** Testing with SILVAMP-LAM was performed according to the manufacturer's instructions using urine from the same aliquot as that used with LF-LAM. The 5-step test procedure is illustrated in an online video [19] and takes 50–60 minutes from start to end result. In brief, urine is added to the reagent tube up to the indicator line (approximately 200 μl), mixed, and incubated for 40 minutes at ambient temperature. After mixing again, 2 drops of urine/reagent are added to the test strip at position 1. Following this, button 2 is pressed immediately. After the "go-next" color indicator mark turns orange (within 3–10 minutes), button 3 is pressed. The result is then read within 10 minutes. The SILVAMP-LAM assay does not use a reference scale card and any visible test line is considered positive.

**Comparator testing (LF-LAM).** LF-LAM is a commercially available lateral flow assay that detects LAM with polyclonal antibodies. It is currently recommended by WHO to assist in the diagnosis of active TB in PLHIV [20]. LF-LAM was used according to the test's package insert. To sum up, 60 μl of urine is applied to the sample pad. After 25 minutes, the test strip is interpreted using the 4-grade reference scale card, with the grade 1 cut-off point as the positivity threshold.

**Blinding.** SILVAMP-LAM was read independently by 2 readers blinded to the results of one another (blinded double reading). After the initial test interpretation, the 2 readers compared results and, in the event of discordance, re-inspected the test to establish a final consensus result (by mutual agreement) that was then used for analysis. The same procedure was used for LF-LAM. SILVAMP-LAM and LF-LAM reading occurred blinded to the test results of the other LAM-based test, patient diagnosis, and all other TB test results. SILVAMP-LAM and LF-LAM results were not available to the assessors of the reference standard.

**Test failure.** In case of SILVAMP-LAM or LF-LAM failure, the test was repeated once. Accuracy calculations were performed from the valid result (first or second attempt).

**Reference standard testing.** For reference standard testing, the specimens were processed using standardized protocols from centralized accredited laboratories of the different sites. The testing flow for each cohort is shown in S2 Table. Sputum, blood, and urine specimens for *Mycobacterium tuberculosis* (*M.tb*) reference standard testing were collected at enrollment,

and additional clinical samples were obtained during hospital admission and at follow-up. Sputum collection across cohorts was done by an experienced nurse or trained clinical research worker, and sputum induction was performed (except for cohort 5) when required. Reference standard testing was performed on all available sputum specimens and included Xpert MTB/RIF (Xpert, Cepheid, Sunnyvale, CA, US; testing predated rollout of Xpert Ultra MTB/RIF), sputum smear microscopy (fluorescence microscopy using Auramine O staining and/or Ziehl–Neelsen staining), Mycobacteria Growth Indicator Tube (MGIT) liquid culture (Becton Dickinson, Franklin Lakes, NJ, US), and solid culture on Löwenstein–Jensen medium. The presence of *M.tb* complex in solid or liquid culture was confirmed with MPT64 antigen detection and/or MTBDR*plus*, MTBC, or CM/AS line probe assays (Bruker Hain [formerly Hain Lifesciences], Nehren, Germany). Blood culture from all participants was done in BAC-TEC Myco/F Lytic culture vials (Becton Dickinson). The exception was that no blood cultures were done for cohorts 4 and 5. WHO-prequalified in vitro diagnostic tests were used for HIV testing (rapid diagnostic tests) and CD4 cell counting (flow cytometry). For urine Xpert testing, 30–40 ml of urine (in cohort 5, only 6 ml) was centrifuged, and, following removal of the supernatant, the pellet was re-suspended in the residual urine volume, then 0.75 ml was tested using Xpert. No urinary Xpert testing was done for cohort 4. For cohorts 2 and 3, additional respiratory and non-respiratory samples such as pleural fluid, cerebrospinal fluid, and tissue fine needle aspirates were obtained, where clinically indicated, and tested using MGIT liquid culture and/or microscopy and/or Xpert. Details of testing according to cohort are reported in S2 Table.

## Case definitions

Patients were assigned to 1 of 4 diagnostic categories using a combination of clinical and laboratory findings. S4 Table indicates the categorization per cohort. Briefly, "definite TB" included patients with microbiologically confirmed *M.tb* (any culture or any Xpert positive for *M.tb*). "Not TB" included patients with all microscopy, culture, and Xpert tests negative for *M.tb* (including at least 1 negative noncontaminated culture result), who were not started on anti-TB treatment and were alive or who improved at 2 to 3 months' follow-up. "Possible TB" was diagnosed in patients who did not satisfy the criteria for "definite TB" but had clinical/radiological features suggestive of TB and were started on TB treatment by non-study clinicians. Patients who did not fall into any of these categories were considered "unclassifiable" and removed from the main analyses but included in a sensitivity analysis. Definition and examples of the "unclassifiable" category can be found in S5 Table.

## Statistical methods

Simple descriptive statistics were used to characterize cohorts. Sensitivity and specificity of the index test were estimated against a microbiological reference standard (MRS) or a composite reference standard (CRS). The "definite TB" and "not TB" categories were used to allocate patients into positive and negative, respectively. The "possible TB" group was considered negative by MRS but positive by CRS, as previously proposed in a study guidance publication [21].

Diagnostic accuracy was determined separately for each cohort, and 95% confidence intervals (95% CIs) were computed using Wilson's score method. Sensitivity and specificity of LF-LAM and SILVAMP-LAM for each cohort were compared using the McNemar test. To estimate sensitivity and specificity across cohorts and CD4 strata, we performed a 2-stage IPD meta-analysis; aggregate data (true positives, false negatives, false positives, true negatives) were extracted from the individual studies and combined using a Bayesian bivariate random-effects model using the meta4diag package [22]. Results are presented with 95% CIs. In a

sensitivity analysis, we assessed the effect on performance when the "unclassifiable" cases were included (a) as MRS negative or (b) as CRS positive. Cohen's kappa coefficient was used to determine inter-reader agreement for SILVAMP-LAM and LF-LAM. The data analysis was performed with R (version 3.5.1) and STATA 15.

# Results

## Study population

Overall, 3,062 potentially eligible participants were screened across the 5 cohorts, of which 1,132 were ineligible according to exclusion criteria predefined in the cohort protocols (Fig 1). HIV-negative participants were excluded and will be reported separately. As a result, 1,930 patients were considered for urine LAM testing on biobanked samples. For the primary analysis, an additional 335 participants were excluded, either due to unavailability of a urine sample ($n = 129$), failed index test ($n = 6$), or being "unclassifiable" ($n = 200$).

Consequently, 1,595 PLHIV across all 5 cohorts were combined for the primary analysis. The majority were inpatients (968; 61%), and 627 (39%) were outpatients. All inpatients came from South African sites, while outpatient data originated primarily from Ghana (63%), with South Africa contributing 28%, and Vietnam the remaining 9%. The characteristics across all PLHIV (and across cohorts) are reported in Table 1.

Participants were typically young adults (median age 37 years [IQR 30–43]), and 59% were female. Forty-three percent had a CD4 count below or equal to 100 cells/μl and 36% above 200 cells/μl. Most participants (94%) had a positive WHO symptom screen for TB upon enrollment. Forty-five percent ($n = 724$) were diagnosed with definite TB, and 119 (7%) died within 2–3 months after enrollment.

## Comparison of diagnostic sensitivity of SILVAMP-LAM and LF-LAM

The meta-analysis of all PLHIV across cohorts showed an overall sensitivity for active TB detection of 70.7% (95% CI 59.0%–80.8%) for SILVAMP-LAM compared to 34.9% (95% CI 19.5%–50.9%) for LF-LAM against the MRS, with a difference of 35.8 percentage points between the 2 tests (Fig 2A). When using the CRS, the sensitivity difference between the assays was 34.4 percentage points: the overall SILVAMP-LAM point estimate was 65.8% (95% CI 55.9%–74.6%) and that of LF-LAM, 31.4% (95% CI 19.1%–43.7%). Amongst inpatients, SILVAMP-LAM sensitivity was 28.1 percentage points higher compared to LF-LAM, and confidence intervals did not overlap (Fig 2B). In outpatients, SILVAMP-LAM sensitivity was 42.7 percentage points higher compared to LF-LAM, but confidence intervals overlapped (Fig 2B). An analysis per cohort, and tables comparing LF-LAM to SILVAMP-LAM results, can be found in S1 Fig, S6 Table and S7 Table.

When patients were stratified by CD4 count, we observed an inverse relationship between sensitivity and CD4 count for both assays (Fig 3A): the sensitivity was higher for lower CD4 counts and systematically decreased for higher CD4 counts. In patients with a CD4 count ≤ 100 cells/μl, SILVAMP-LAM had a sensitivity of 87.1% (95% CI 79.3%–93.6%), compared to 56.0% (95% CI 43.9%–64.9%) for LF-LAM (Fig 3A). A similar difference in sensitivity was observed in patients with less severe immunosuppression (CD4 > 200 cells/μl), but overall sensitivity was lower for both assays in this group: 43.9% (95% CI 34.3%–53.9%) for SILVAMP-LAM and 10.9% (95% CI 5.2%–18.4%) for LF-LAM (Fig 3A). The difference in performance in inpatients versus outpatients was largely explained by the differences in the distribution of the populations across CD4 strata, as outlined in Fig 3B.

Additional information on the distribution of patients across cohorts by CD4 group and smear status is provided in S8 Table and S9 Table. An analysis of accuracy by smear status is

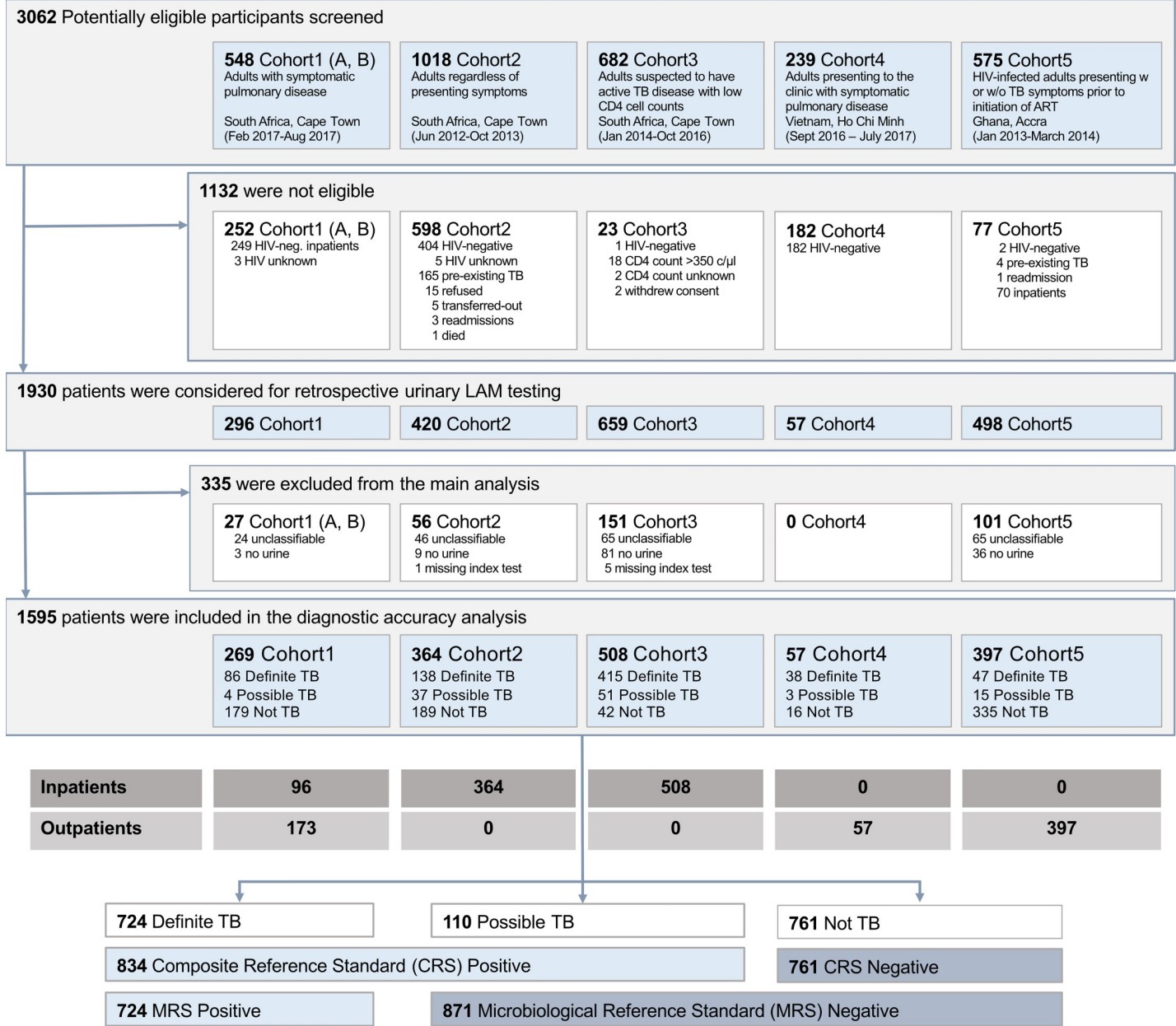

**Fig 1. Flow diagram showing the study populations and the number of patients included overall, per cohort, per hospitalization status, and per TB case definition.** CRS, composite reference standard; LAM, lipoarabinomannan; MRS, microbiological reference standard; TB, tuberculosis; w or w/o, with or without.

provided in S2 Fig. A sensitivity analysis including those falling in the "unclassifiable" category is reported in S10 Table. When the "unclassifiable" cases were included as CRS positive, the sensitivity decreased for both tests, but more for SILVAMP-LAM (12.2%) than for LF-LAM (4.7%).

## Comparison of diagnostic specificity of SILVAMP-LAM and LF-LAM

The specificity estimates, when using the MRS for the meta-analysis, were 90.9% (95% CI 87.2%–93.7%) and 95.3% (95% CI 92.2%–97.7%) for SILVAMP-LAM and LF-LAM,

**Table 1. Demographic and clinical characteristics of all PLHIV.**

| Characteristic | All PLHIV (*n* = 1,595) | Inpatient PLHIV (*n* = 968) | Outpatient PLHIV (*n* = 627) |
|---|---|---|---|
| **Demographic and clinical characteristics** | | | |
| Age—years | 36.6 (30; 43) | 35 (30; 42) | 38 (31; 44) |
| Female sex | 939 (59%) | 523 (54%) | 416 (66%) |
| Positive WHO TB symptom screen | 1,498 (94%) | 933 (96%) | 565 (90%) |
| History of TB | 536 (34%) | 439 (45%) | 97 (16%) |
| Antiretroviral therapy | 558 (35%) | 394 (41%) | 164 (26%) |
| CD4 count—cells/μl | 126 (42; 302) | 86 (33; 190) | 249 (91; 477) |
| **Distribution in diagnostic categories** | | | |
| Definite TB | 724 (45%) | 600 (62%) | 124 (20%) |
| Possible TB | 110 (7%) | 91 (9%) | 19 (3%) |
| Not TB | 761 (48%) | 277 (29%) | 484 (77%) |
| **CD4 count (cells/μl)** | | | |
| 0 to 100 | 677 (43%) | 516 (53%) | 161 (26%) |
| 101 to 200 | 319 (20%) | 216 (22%) | 103 (16%) |
| >200 | 581 (36%) | 231 (24%) | 350 (56%) |
| Unknown | 18 (1%) | 5 (1%) | 13 (2%) |
| **Outcome** | | | |
| Died within 2–3 months | 119 (7%) | 105 (11%) | 14 (2%) |
| Alive | 1,354 (85%) | 810 (84%) | 544 (87%) |
| Lost to follow-up | 35 (2%) | 16 (2%) | 19 (3%) |
| No follow-up done/required* | 87 (6%) | 37 (4%) | 50 (8%) |

Data are given as median (interquartile range) or number (percent).

*No follow-up done/required: empty follow-up field in case report form of cohort 4 or no follow-up required at other sites.

PLHIV, people living with HIV; TB, tuberculosis; WHO, World Health Organization.

respectively (Fig 2A); however, the confidence intervals overlapped. When using the CRS, the overall specificity estimates were higher: 93.4% (95% CI 89.3%–96.2%) for SILVAMP-LAM and 97.3% (95% CI 95.1%–98.9%) for LF-LAM (Fig 2A). Here too, the confidence intervals overlapped. The specificity of SILVAMP-LAM was lower amongst those with CD4 ≤ 100 cells/μl (80.5% using the MRS and 85.2% using the CRS) compared to those in the higher CD4 count strata (Fig 3A). Of the 47 patients with SILVAMP-LAM false-positive results, 34 (72%) came from patients with CD4 count ≤ 100 cells/μl, while overall this CD4 group included 43% of patients. Nontuberculous mycobacteria (NTM) were cultured from sputum (*M. avium/intracellulare* complex) in 2 of the patients classified as "not TB" but positive by SILVAMP-LAM. A more detailed analysis of all false-positive results, with additional information on the clinical and laboratory workup of the individual patients, is provided in S11 Table.

When the "unclassifiable" patients were included in a sensitivity analysis as MRS negative, the specificity remained largely the same (with a difference of −0.4 percentage points for SILVAMP-LAM and −0.8 percentage points for LF-LAM).

## Invalid results and inter-reader agreement

Out of 1,801 initial test runs with SILVAMP-LAM, 26 tests (1.4%) failed (by comparison to 6 that failed for LF-LAM; 0.3%). Reasons for failure are reported in S12 Table. The most common reason for failure was that no control line was present (12/26). A repeat was possible for 23 out of 26 samples (3 had insufficient sample), and 20 yielded a valid result on repeat. Thus,

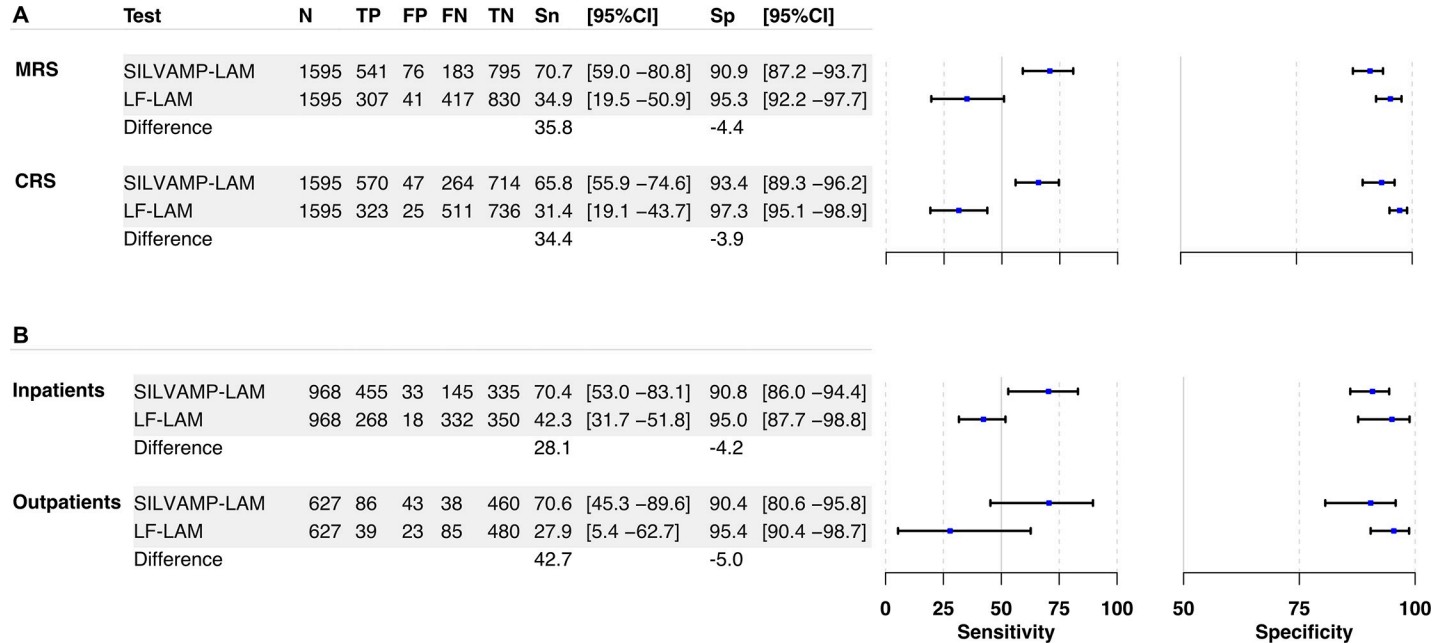

**Fig 2. Accuracy of SILVAMP-LAM and LF-LAM against different reference standards and in in- and outpatients.** Forest plots of sensitivity and specificity and differences between SILVAMP-LAM and LF-LAM (A) against the microbiological and composite reference standards for all cohorts combined and (B) against the MRS by in- and outpatients. The axis for sensitivity ranges from 0% to 100%, while the axis for specificity ranges from 50% to 100%. CRS, composite reference standard; FN, false negative; FP, false positive; LF-LAM, Alere Determine TB LAM lateral flow assay; MRS, microbiological reference standard; SILVAMP-LAM, Fujifilm SILVAMP TB LAM; Sn, sensitivity; Sp, specificity; TN, true negative; TP, true positive.

there was an overall invalid rate of 0.16% (3/1,824 runs). Interrater agreement was very high both for SILVAMP-LAM and LF-LAM (kappa 0.95 and 0.92, respectively; S13 Table).

## Discussion

In this assessment of 1,595 hospitalized and non-hospitalized PLHIV from settings with high TB burden, the SILVAMP-LAM assay identified a substantially higher proportion of TB patients than LF-LAM. While the specificity was lower for SILVAMP-LAM than for LF-LAM, the 95% confidence intervals of specificity for these tests overlapped. In all sub-analyses by CD4 count (Fig 3), the sensitivity of SILVAMP-LAM was higher (range 23.8–37.4 percentage points) than that of LF-LAM, with non-overlapping confidence intervals for the inpatient cohort (likely due to the higher case number, which allowed more accurate estimates).

As in previous reports [11,23], we found an inverse relationship between the sensitivity of LAM tests and CD4 count (Fig 3), and sensitivities of the tests were lowest in cohort 1B, where the majority of patients had CD4 count > 200 cells/μl (S1 Fig; S9 Table). The sensitivity of SIL-VAMP-LAM was the highest (87.1%, 95% CI 79.3%–93.6%) in patients with the highest risk of having disseminated TB and of death from TB associated with severe immunosuppression (patients with CD4 ≤ 100 cells/μl) [24]. SILVAMP-LAM could have diagnosed TB in up to 89% of patients who died [25], and testing, particularly in immunocompromised patient populations, is expected to have a mortality benefit, as has been shown for LF-LAM [2,3], although this still needs to be evaluated. The moderate sensitivity that was maintained even in the highest CD4 stratum (44.4%) will result in a higher diagnostic yield for SILVAMP-LAM and will offer an expanded opportunity of rapid diagnosis on presentation in this patient population.

The increase in sensitivity that had been observed in inpatients was confirmed in outpatients, and differences in performance of SILVAMP-LAM between in- and outpatients were

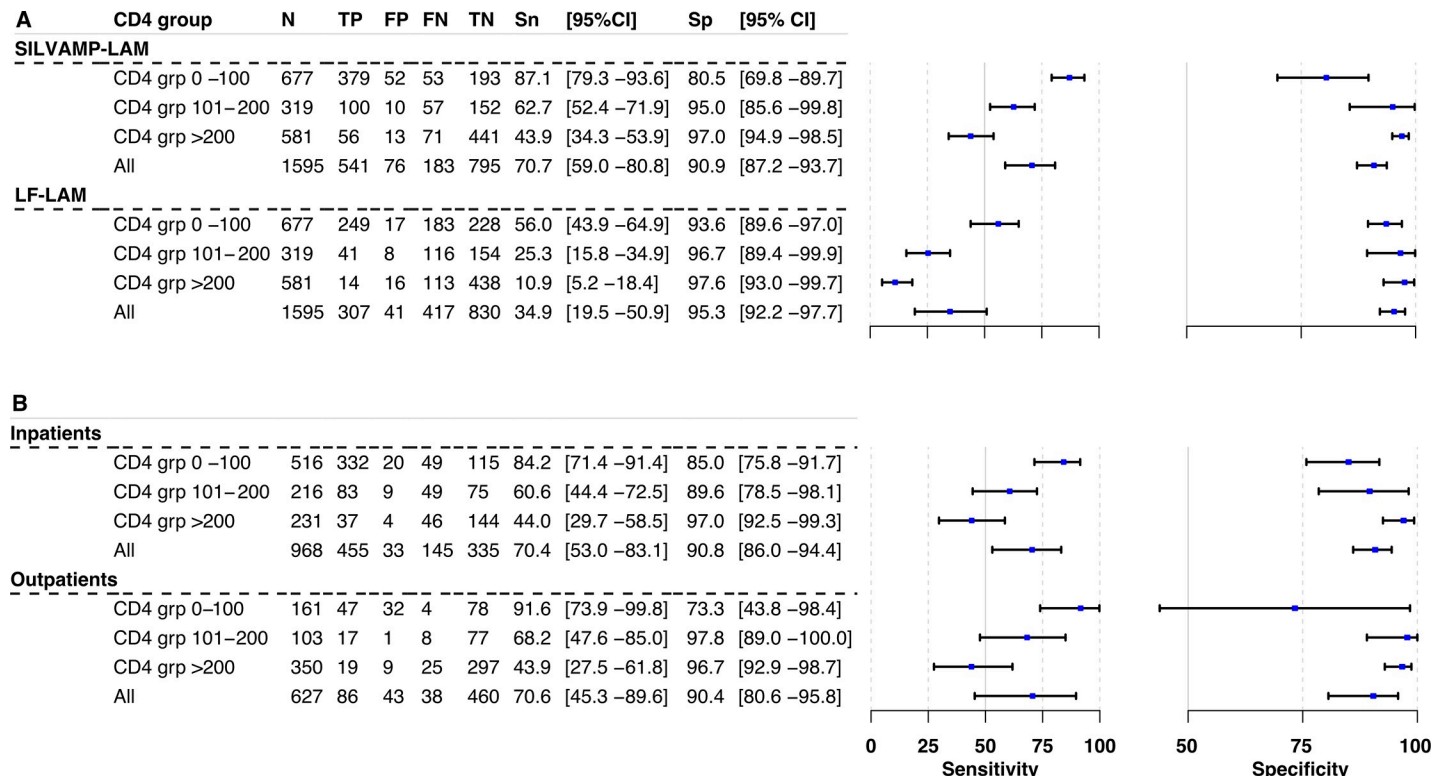

**Fig 3. Accuracy of SILVAMP-LAM and LF-LAM across CD4 strata.** Forest plots of sensitivity and specificity of (A) SILVAMP-LAM and LF-LAM in CD4 strata ≤100, 101–200, and >200 cells/µl against the microbiological reference standard for all cohorts combined and (B) SILVAMP-LAM only against the microbiological reference standard in in- and outpatients in the CD4 strata. The axis for sensitivity ranges from 0% to 100%, while the axis for specificity ranges from 50% to 100%. FN, false negative; FP, false positive; LF-LAM, Alere Determine TB LAM lateral flow assay; SILVAMP-LAM, Fujifilm SILVAMP TB LAM; TN, true negative; TP, true positive.

explained by differences in CD4 group distributions [11]. For LF-LAM, the CD4 count distribution did not fully explain the sensitivity difference observed between in- and outpatients. This finding is in line with the results published in the updated Cochrane meta-analysis for LF-LAM [23].

The point estimates of specificity for SILVAMP-LAM were lower than those for LF-LAM using both the MRS and the CRS. However, the lower specificity of both LF-LAM and SILVAMP-LAM could be explained in part by an imperfect reference standard that lacks complete sensitivity (i.e., reference-standard-negative, LAM-positive results that represent true TB) [26]. It is possible that an imperfect reference standard could disproportionally affect a more sensitive test. The lower specificity seen with a decreased CD4 count in this study, and the improved specificity seen with the CRS compared to the MRS, further support this explanation. Also, in higher CD4 count strata, the specificities of the 2 tests approximate each other, which provides further supporting evidence. In addition, cohorts with no or with only limited culture or Xpert testing in blood or urine (S2 Table and S1 Fig; cohorts 4 and 5) reported lower specificities, which again might point towards underdiagnosis by the reference standard, as shown in previous studies [26,27].

An alternative explanation for the reduced specificity of SILVAMP-LAM could be cross-reactivity to other pathogens. This was a problem with LF-LAM, as the polyclonal antibodies used in the test are known to react with urinary tract pathogens and fast-growing NTM [28,29]. Cross-reactivity to common urinary tract pathogens and fast-growing NTM has been excluded in studies assessing the antibodies used in SILVAMP-LAM, but some cross-reactivity

has been observed with slow-growing NTM [28,30]. Our data suggest that cross-reactivity is a small problem, if at all, as slow-growing NTM were observed in only 2 out of 47 patients with false-positive results on SILVAMP-LAM (S11 Table).

This meta-analysis has some limitations, pointing to areas needing further research. SILVAMP-LAM testing was done using biobanked specimens from hospitalized and non-hospitalized patients. However, there is evidence to suggest that data from retrospective LAM testing based on frozen samples are comparable to data from testing on fresh samples [31,32]. The operators in the current study were highly skilled laboratory personnel in reference centers. SILVAMP-LAM may have the potential to be implemented as a POC assay in primary care clinics, HIV clinics, or TB microscopy centers. The accuracy, feasibility, and acceptability in these settings, with often less-skilled workers, needs to be evaluated in a separate study, particularly considering the slightly higher complexity of SILVAMP-LAM over LF-LAM. The inclusion of different cohorts with different study designs resulted in heterogeneity and different exclusion rates; however, differences in diagnostic sensitivity between SILVAMP-LAM and LF-LAM persisted in a sensitivity analysis including the "unclassifiable" category (S10 Table) and in the analyses by CD4 subgroup (Fig 3). Further, antiretroviral treatment may have influenced patient outcome and thus the reference standard categories "possible TB" and "unclassifiable."

SILVAMP-LAM has the potential to have similar or improved favorable effects on patient outcomes compared to LF-LAM, although a prospective study in relevant clinical settings is needed to evaluate this.

Neither LF-LAM nor SILVAMP-LAM can differentiate drug-resistant from drug-sensitive TB, and therefore it is important that these rapid diagnostic tools are supplemented with drug susceptibility testing. Evaluation of SILVAMP-LAM in diagnostic algorithms therefore should be considered.

In 2019, over 5 years after the establishment of the WHO policy on LF-LAM, a survey of 31 countries with high TB/HIV burden, with responses obtained from 24, showed that only 11 countries had LF-LAM policies in place, with only 5 countries currently using LF-LAM [10]. Limited budgets, lack of country-specific data, administrative hurdles such as local regulatory approval, lack of coordination between national TB and HIV programs, and small perceived patient population size were the most commonly cited constraints on LF-LAM adoption [10]. There is real potential for a broader and simpler WHO recommendation for SILVAMP-LAM given its higher sensitivity, and this could help to overcome some of these implementation barriers.

Collectively, these results suggest that, if implemented in clinical practice and linked with appropriate treatment, the SILVAMP-LAM assay would allow for earlier diagnosis of HIV-associated TB in a larger proportion of hospitalized and non-hospitalized patients compared to the current LF-LAM test.

## Supporting information

**S1 Data.**
(XLSX)

**S1 Fig. Forest plots of sensitivity and specificity for SILVAMP-LAM and LF-LAM by cohort against the microbiological reference standard (MRS) and the composite reference standard (CRS).**
(TIFF)

**S2 Fig. Accuracy by smear status.**
(TIFF)

**S1 PRISMA IPD Checklist.**
(DOCX)

**S1 STARD Checklist.**
(DOCX)

**S1 Table. Study population, setting and location, eligibility, and inclusion and exclusion criteria used for the studies.**
(DOCX)

**S2 Table. Specimen collection and testing flow by cohort.**
(DOCX)

**S3 Table. Specimen storage.**
(DOCX)

**S4 Table. Diagnostic categories by cohort.**
(DOCX)

**S5 Table. Definition and examples of "unclassifiable" category.**
(DOCX)

**S6 Table. Two-by-two table of SILVAMP-LAM versus LF-LAM among "definite TB" patients.**
(DOCX)

**S7 Table. Two-by-two table of SILVAMP-LAM versus LF-LAM among "not TB" patients.**
(DOCX)

**S8 Table. Analysis by cohort, smear status, and CD4 group for all HIV-positive inpatients.**
(DOCX)

**S9 Table. Analysis by cohort, smear status, and CD4 group for all HIV-positive outpatients.**
(DOCX)

**S10 Table. Sensitivity analysis of diagnostic accuracy for all PLHIV (including "unclassifiable" patients) by MRS and CRS.**
(DOCX)

**S11 Table. Further Information on patients categorized as "not TB" with positive SILVAMP-LAM results.**
(DOCX)

**S12 Table. SILVAMP-LAM failure rates and errors for all samples tested.**
(DOCX)

**S13 Table. Agreement of 2 independent test readers for all samples tested.**
(DOCX)

**S1 Translation. Japanese translation of the abstract by Satoshi Mitarai.**
(DOCX)

## Acknowledgments

The authors thank the late Stephen D. Lawn, who designed and led the cohort 2 study.

## Author Contributions

**Conceptualization:** Tobias Broger, Claudia M. Denkinger, Samuel G. Schumacher.

**Data curation:** Tobias Broger, Rita Székely, Stephanie Bjerrum, Charlotte Schutz, Japheth A. Opintan, Satoshi Mitarai, Kinuyo Chikamatsu, Andrew D. Kerkhoff, Aurélien Macé.

**Formal analysis:** Tobias Broger, Mark P. Nicol, Rita Székely, Stephanie Bjerrum, Bianca Sossen, Charlotte Schutz, Japheth A. Opintan, Isik S. Johansen, Satoshi Mitarai, Kinuyo Chikamatsu, Andrew D. Kerkhoff, Aurélien Macé, Stefano Ongarello, Graeme Meintjes, Claudia M. Denkinger, Samuel G. Schumacher.

**Funding acquisition:** Tobias Broger, Mark P. Nicol, Stephanie Bjerrum, Graeme Meintjes, Claudia M. Denkinger.

**Investigation:** Bianca Sossen, Satoshi Mitarai.

**Methodology:** Tobias Broger, Mark P. Nicol, Bianca Sossen, Andrew D. Kerkhoff, Aurélien Macé, Graeme Meintjes, Claudia M. Denkinger, Samuel G. Schumacher.

**Project administration:** Tobias Broger, Rita Székely, Bianca Sossen, Kinuyo Chikamatsu.

**Supervision:** Mark P. Nicol, Satoshi Mitarai, Stefano Ongarello, Graeme Meintjes, Claudia M. Denkinger.

**Validation:** Mark P. Nicol, Stephanie Bjerrum, Charlotte Schutz, Japheth A. Opintan, Isik S. Johansen, Satoshi Mitarai, Kinuyo Chikamatsu, Andrew D. Kerkhoff, Aurélien Macé, Graeme Meintjes, Claudia M. Denkinger, Samuel G. Schumacher.

**Visualization:** Tobias Broger, Mark P. Nicol, Stephanie Bjerrum, Bianca Sossen, Aurélien Macé, Stefano Ongarello, Claudia M. Denkinger, Samuel G. Schumacher.

**Writing – original draft:** Tobias Broger, Mark P. Nicol, Rita Székely, Stephanie Bjerrum, Bianca Sossen, Aurélien Macé, Stefano Ongarello, Claudia M. Denkinger, Samuel G. Schumacher.

**Writing – review & editing:** Tobias Broger, Mark P. Nicol, Rita Székely, Stephanie Bjerrum, Bianca Sossen, Charlotte Schutz, Japheth A. Opintan, Isik S. Johansen, Satoshi Mitarai, Kinuyo Chikamatsu, Andrew D. Kerkhoff, Aurélien Macé, Stefano Ongarello, Graeme Meintjes, Claudia M. Denkinger, Samuel G. Schumacher.

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
