## [Decision Letter · Decision Letter 0]

31 Jan 2020

Dear Dr. Denkinger,

Thank you very much for submitting your manuscript "Diagnostic accuracy of a novel point-of-care urine lipoarabinomannan assay for people living with HIV – a meta-analysis of in- and outpatient data" (PMEDICINE-D-19-03864) for consideration at PLOS Medicine. 

Your paper was evaluated by an academic editor with relevant expertise, and sent to independent reviewers, including a statistical reviewer. The reviews are appended at the bottom of this email and any accompanying reviewer attachments can be seen via the link below:

[LINK]

In light of these reviews, we will not be able to accept the manuscript for publication in the journal in its current form, but we would like to invite you to submit a revised version that fully addresses the reviewers' and editors' comments. You will appreciate that we cannot make a decision about publication until we have seen the revised manuscript and your response, and we expect to seek re-review by one or more of the reviewers. 

We hope to receive your revised manuscript by Feb 21 2020 11:59PM. Please email us (plosmedicine@plos.org) if you have any questions or concerns.

Please let me know if you have any questions. Otherwise, we look forward to receiving your revised manuscript in due course. 

Sincerely,

Richard Turner PhD, for Thomas McBride, PhD

rturner@plos.org

Our academic editor commented: "Using brands in scientific articles is something it is better to avoid. Could the authors instead specify the technology used in each assay to differentiate the two?"

We suggest amending the data statement (to be presented in the metadata in the event of publication) to "Yes - all data available" reflecting that the data for the analyses presented in the current study are available. Please include these data in the form of supplementary files or, if this is not possible, supply a non-author contact for readers interested in inquiring about data access. Please let me know if it would be helpful to discuss this further. 

Please substitute a colon in your title after "HIV".

At line 78, please start the sentence "In this study, we found that ..." or similar. 

In your abstract and elsewhere in the paper, please quote p values alongside 95% CI where available. 

After your abstract, we will need to ask you to add a new and accessible "author summary" section in non-identical prose. You may find it helpful to consult one or two recent research papers published in PLOS Medicine to get a sense of the preferred style. 

Early in the methods section of your main text, please state whether the current analysis had a specific protocol or prespecified analysis plan, and if so attach the document(s) as a supplementary file (referred to in the text). Please highlight analyses that were not prespecified. 

Please update the reference at line 408. 

Throughout the text, please format reference call-outs as follows: "... immunocompromised PLHIV [7,8], ...".

Please remove trade marks throughout your text. 

Please spell out the group author name for references 9, 17 and 18.

Please add full access details for references 11 and 23. 

Please ensure that all journal names are abbreviated as appropriate in your reference list, e.g., reference 19.

We ask you to adapt your PRISMA checklist so that individual items are referred to by section (e.g., "Methods") and paragraph number rather than by page or line numbers, as the latter generally change in the event of publication. 

Please use the PRISMA IPD document (http://www.prisma-statement.org/documents/PRISMA%20IPD%20checklist.pdf). 

Please refer to the document in the methods section, around line 156.

Comments from the reviewers:

*** Reviewer #1: 

[See attachment]

Michael Dewey

*** Reviewer #2: 

A very interesting study with several strengths:

1. Clear and well written

2. Findings are important. The new Fuji test appears to be a major improvement over the old Alere test.

3. Multiple cohorts from different settings strengthens the findings

4. Stratified analyses are useful to position the findings.

Weaknesses/major comments:

1. My major concern revolves around the practical utility of the test. The sensitivity is adequate only in persons with advanced HIV. In HIV infected with CD4 >100 sensitivity drops, and if CD4>200 the test is of questionable clinical utility. We need to know more about when the test functions best, and when it is sub-optimal. A major piece of information, that I could not find anywhere (which is puzzling - not in Tables, nor text) is ARV treatment. How many were on ARV, at time of testing, how many were started within 2 weeks, or 1 month, etc - this would influence the clinical reference standard a lot. 

2. I strongly recommend the addition of the findings from the HIV negative participants in the different cohorts. This will help inform what settings this test will be useful to implement. If sensitivity is very low in HIV negative, this wil be useful to decision makers whether to invest in this test.

3. I suggest the sensitivity/specificity at higher cut-points, or thresholds of CD4 (>300, >400, etc)

4. The authors acknowledge that for the study the test was conducted by skilled technicians. Can they provide some information on initial and on-going training and supervision required. And, indeed, what were the qualifications of the technicians involved. This is an often overlooked point in diagnostic studies. Reading the description, the test seems a little tricky, so more info on the training and qualifications of the techs involved would be useful to be able to gauge whether this really can be a POC test.

5. Results - abstract: The authors mention the stratified analysis that favours the test (CD4<100). They should also mention the very important finding of 63% sensitivity if CD4 100-200, and 44% if CD>200.

6. Conclusions - abstract: I find some of the conclusions, especially in the abstract, not supported by the data presented. "In patients with the greatest risk of death from TB (patients with CD4 ≤100 cells/μl), the sensitivity of FujiLAM was highest. Thus, FujiLAM is expected to have at least a similar mortality benefit to that previously shown for AlereLAM". I did not see results of sensitivity or specificity among those who died vs survived, so I suggest they simply state what they did show - sensitivity was greatest if CD4<100. The "expected mortality benefit" is purely speculative. Similarly, "and to establish whether reduced specificity may be due to missed detection of TB by other diagnostic tests currently in use" This seems speculative to me; they did see that when the clinical standard truly excluded TB (although the requirement for improvement may have selected for the more healthy/ milder disease) the specificity was higher - this finding can be mentioned. 

Minor points:

1. I find the data access to be not in conformity with PLOS Med policy. 

2. The COI statement reads more like publicity for FIND. More importantly, the nature of the agreement between FIND and the test developers is unclear and must be clarified in terms of financial considerations, ownership of rights to the test, and future access to the test. The comments I have made about the abstract results and conclusions reflect a concern over potential bias. No doubt the investigators are enthusiastic about this test; this 'optimism bias' afflicts us all. But this makes it all the more important to understand the exact financial relationship between FIND and the test developers. And to guarantee public access to the data. (ie these are related concerns)

*** Reviewer #3: 

This is a very well written individual patient data level meta-analysis of the diagnostic accuracy of FujiLAM, with a comparison to AlereLAM. The data presented are important in the field of HIV-associated TB, and I believe the methodology used is appropriate. My only major concern is the overlap between this article, and the original manuscript published in the Lancet ID (ref 11 in the manuscript) which already combines the inpatient data from cohorts 1-3 (includes all the in-patient data presented in this manuscript). I do not think the authors have adequately acknowledged or justified this duplication in the current manuscript, and why this manuscript did not simply present the outpatient data. The discussion and conclusions are very similar between both papers. 

Minor points

-Line 90, I disagree with the statement that MOST deaths would be preventable if TB were diagnosed earlier, the evidence to support this statement does not exist

-Line 106 mentions low programmatic uptake of AlereLAM, could the authors add to the discussion if/why they think uptake of FujiLAM might be better

-Please add heterogeneity of reference TB tests in the different studies to the limitations in the discussion section

-Lines 390-394, please clarify that the CIs for outpatients did overlap

-Please add to appendix details of how urine specimens were biobanked, including temperature and duration etc, as this could impact accuracy

***

[LINK]

---

## [Decision Letter · Decision Letter 1]

20 Mar 2020

Dear Dr. Denkinger,

Thank you very much for re-submitting your manuscript "Diagnostic accuracy of a novel tuberculosis point-of-care urine lipoarabinomannan assay for people living with HIV: a meta-analysis of individual in- and outpatient data" (PMEDICINE-D-19-03864R1) for consideration at PLOS Medicine.

I have discussed the paper with our academic editor and it was also seen again by two reviewers. I am pleased to tell you that, provided the remaining editorial and production issues are dealt with, we expect to be able to accept the paper for publication in the journal.

[LINK]

Please let me know if you have any questions. Otherwise, we look forward to receiving the revised manuscript shortly. 

Kind regards,

Richard Turner, PhD

rturner@plos.org

Requests from Editors:

Please note in your competing interests statement that CMD is a member of PLOS Medicine's Editorial Board. 

Noting your first point of response, please do adopt the terminology "LfLAM" and "LfSilvAmpLAM" throughout the paper. 

Early in the "methods and findings" subsection of your abstract, please quote the range of years over which the studies were conducted (e.g., "... and Ghana, carried out during 2012-17"). 

At line 82, "... proportion of TB cases"?

In the abstract and throughout the paper, please provide p values alongside 95% CI where available. 

You mention twice that the meta-analysis follows PRISMA guidelines (lines 155 and 197), and once will suffice. Please refer to the attached checklist (e.g., "see S1_PRISMA_Checklist") at this mention. 

In your reference list, please remove the academic editors' names from references 4 and 30, and any others. 

Please remove "Lancet Infectious Diseases" from reference 19 (we believe the relevant paper is reference 11) and substitute "YouTube". 

We ask you to also provide a completed STARD checklist as a supplementary document, again referred to in your methods section. 

Comments from academic editor:

(1) I do think we should avoid brand names and refer to the technology. This is what we always do for medicines so think it is appropriate for diagnostics as well. Using LfLAM for the assay that relies on lateral flow alone and LfSilvAmpLAM to refer to the assay that uses both lateral flow and silver amplification seems to be a good alternative. 

(2) The highest risk for misinterpretation occurs when p-values are presented without 95% CIs. I understand the authors' concerns about p-values, but we are asking that they provide both the 95% CI and p-value. 

(3) The authors should state that the data for HIV negative persons are being analysed and will be published separately somewhere in the MS (this was in their response but did not see it the text)?

(4) Similarly, the issues with performance across CD4 strata and with respect to ART status should be crystal clear in the abstract and the main text.

Comments from Reviewers:

*** Reviewer #1: 

The authors have met my points.

I have just one minor comment. I would leave in the statement about the use of the die for selection, it is what was done and although perhaps sub-optimal it could not be called wrong in the field.

Michael Dewey

*** Reviewer #3: 

I am happy that all the reviewers comments have been adequately addressed

***

[LINK]

---

## [Editor Report · Decision Letter 2]

9 Apr 2020

Dear Dr. Denkinger, 

On behalf of my colleagues and the academic editor, Dr. Amitabh Bipin Suthar, I am delighted to inform you that your manuscript entitled "Diagnostic accuracy of a novel tuberculosis point-of-care urine lipoarabinomannan assay for people living with HIV: a meta-analysis of individual in- and outpatient data" (PMEDICINE-D-19-03864R2) has been accepted for publication in PLOS Medicine. 

PRODUCTION PROCESS

PRESS

PROFILE INFORMATION

Thank you again for submitting the manuscript to PLOS Medicine. We look forward to publishing it. 

Best wishes, 

Richard Turner, PhD

Senior Editor 

PLOS Medicine

plosmedicine.org